# Micro- and Mycobiota Dysbiosis in Pancreatic Ductal Adenocarcinoma Development

**DOI:** 10.3390/cancers13143431

**Published:** 2021-07-08

**Authors:** Ruben Bellotti, Cornelia Speth, Timon E. Adolph, Cornelia Lass-Flörl, Maria Effenberger, Dietmar Öfner, Manuel Maglione

**Affiliations:** 1Department of Visceral, Transplant and Thoracic Surgery, Center of Operative Medicine, Medical University of Innsbruck, 6020 Innsbruck, Austria; ruben.bellotti@tirol-kliniken.at (R.B.); dietmar.oefner@i-med.ac.at (D.Ö.); 2Institute of Hygiene and Medical Microbiology, Medical University of Innsbruck, 6020 Innsbruck, Austria; cornelia.speth@i-med.ac.at (C.S.); cornelia.lass-floerl@tirol-kliniken.at (C.L.-F.); 3Department of Internal Medicine I, Gastroenterology, Hepatology, Metabolism & Endocrinology, Medical University of Innsbruck, 6020 Innsbruck, Austria; timon-erik.adolph@i-med.ac.at (T.E.A.); maria.effenberger@tirol-kliniken.at (M.E.)

**Keywords:** pancreatic cancer, microbiome, mycobiome, inflammation, immunosuppression, tumor initiation, tumor progression, *Proteobacteria*, *Malassezia*

## Abstract

**Simple Summary:**

Pancreatic ductal adenocarcinoma (PDAC) is currently the third leading cause of cancer-related mortality. Still, screening diagnostic, carcinogenesis and therapeutic strategies are a matter of debate. Recent research on PDAC focused on the microbial community residing in the pancreas which is formed by bacteria (microbiota) and fungi (mycobiota). These microorganisms that are associated with different pancreatic pathologies reveal fascinating, new research frontiers. Specific microbial signatures may arise as novel screening tools for early diagnosis. Local or distant effects of microbes could reveal themselves as the missing link between immunological dysregulation and PDAC initiation and/or progression. Most importantly, micro- and mycobiota may represent a promising target for multimodal treatment concepts in a (neo)adjuvant or even in a tumor prevention setting. Herein we present a review of the current literature proposing a model of how the micro- and the mycobiome may be intertwined with PDAC occurrence.

**Abstract:**

Background: Dysbiosis of the intestinal flora has emerged as an oncogenic contributor in different malignancies. Recent findings suggest a crucial tumor-promoting role of micro- and mycobiome alterations also in the development of pancreatic ductal adenocarcinoma (PDAC). Methods: To summarize the current knowledge about this topic, a systematic literature search of articles published until October 2020 was performed in MEDLINE (PubMed). Results: An increasing number of publications describe associations between bacterial and fungal species and PDAC development. Despite the high inter-individual variability of the commensal flora, some studies identify specific microbial signatures in PDAC patients, including oral commensals like *Porphyromonas gingivalis* and *Fusobacterium nucleatum* or Gram-negative bacteria like *Proteobacteria*. The role of *Helicobacter* spp. remains unclear. Recent isolation of *Malassezia globosa* from PDAC tissue suggest also the mycobiota as a crucial player of tumorigenesis. Based on described molecular mechanisms and interactions between the pancreatic tissue and the immune system this review proposes a model of how the micro- and the mycobial dysbiosis could contribute to tumorigenesis in PDAC. Conclusions: The presence of micro- and mycobial dysbiosis in pancreatic tumor tissue opens a fascinating perspective on PDAC oncogenesis. Further studies will pave the way for novel tumor markers and treatment strategies.

## 1. Introduction

Pancreatic ductal adenocarcinoma (PDAC) is currently the third leading cause of cancer-related mortality. About 90% of PDAC cases are diagnosed in patients older than 55 years, and with increased longevity in the general population PDAC burden is expected to rise. Still, the survival is abysmal, with a 5-year survival rate of 8.2% [1].

Surgical resection is currently considered to be the only curative treatment. However, only 15% of patients present with a resectable disease at diagnosis [1]. Moreover, patients undergoing resection and adjuvant chemotherapy have a limited prognosis with a median overall survival between 28 and 54 months [2,3,4].

PDAC occurs mostly sporadically. Risk factors include smoking, heavy alcohol intake, history of chronic pancreatitis (CP), overweight, and diabetes. Notably, many of these factors are related to alterations of the gut flora [5,6]. This microbial community presents a wide inter-individual variability, depending on host-specific factors, such as age, gender, genotype, and bile acids production [7]. Of note, it has been shown that the pancreas can also shape the flora by specific antimicrobial peptide secretion [8].

The intestinal micro- and mycobiome have recently gained increasing interest in the field of PDAC with studies suggesting a tumorigenic relevance of both bacterial and fungal dysbiosis. This review aims to give an overview of the alteration patterns of bacterial and fungal flora associated with PDAC and to highlight possible molecular pathways linking bacterial and fungal dysbiosis with pancreatic carcinogenesis.

## 2. Methods

A systematic PubMed/MEDLINE literature search was performed for PDAC related alterations of bacterial and fungal flora. Keywords included “pancreatic ductal adenocarcinoma”, in combination with “microbiome”, “microbiota” “mycobiome”, “fungi” or “dysbiosis” and “risk-factor”, “pilot-study” or “systematic review”.

## 3. Microbiome Alterations and PDAC

The human intestine bears more than 5.000 different bacterial species (10^14^ microorganisms), which are fundamental for regulating the balance between health and disease [9]. Microbial dysbiosis and disrupted epithelial barriers can promote bacterial translocation favoring neoplastic transformation. The microbiome’s oncogenesis contribution has emerged in different malignancies of the gastrointestinal tract, such as esophageal, gastric and colorectal carcinoma [9,10]. However, only few studies associate the gut flora with tumor development of non-gastrointestinal tract tissues. One example is PDAC [9,11].

### 3.1. Alterations of the Oral Microbiome and PDAC

Alterations of the oral microbiota have been linked to PDAC in different studies. In periodontal disease, pathogenetic oral flora and tooth loss have been described as independent risk factors for PDAC development [12,13,14]. The characterization of a specific oral microbiome-shift in some group of patients showed higher levels of *Aggregatibacter actinomycetemcomitans*, *Bacteroides* spp., *Granulicatella adiacens*, and *Porphyromonas gingivalis*. Consistent presence of *P. gingivalis* and *Aggregatibacter* spp. was predictive for PDAC development (even when detected many years before PDAC development) [15], and correspondent plasma antibodies were elevated in PDAC subjects [16]. Of note in this context, *P. gingivalis* has been demonstrated to be able to survive both inside human and murine pancreatic cancer cells in vitro, especially under hypoxic conditions, which is a typical trait of PDAC [17].

On the contrary, some other bacterial taxa like *Veillonella* spp. and *Neisseria elongata* have been found to be negatively associated with PDAC, thus representing a possible protective factor against this type of malignancy [15,18,19].

Curiously, contrasting results come from the analysis of *Streptococcus*, *Fusobacterium nucleatum* and *Leptorichia*. While the carrier-status related to these taxa was described by some groups to be associated with a decreased risk of developing PDAC [15,19,20], others found a positive correlation [18,21]. Interestingly, one study showed higher serum and salivary antibodies against *F. nucleatum* in patients with high-grade dysplasia intraductal papillary mucinous neoplasms (IPMN) or IPMN with associated invasive cancer compared to patients bearing a low-risk IPMN [22]. However, other studies could not confirm these differences [23].

### 3.2. The Presence of Helicobacter and PDAC

Results on the correlation between *Helicobacter* spp. and PDAC are inconsistent. Serological analysis showed in some studies a positive correlation between *H. pylori* and PDAC [13,24,25,26], however, CagA-positive strains of *H. pylori* showed no significant association with pancreatic cancer [16,27,28,29,30]. In this sense, an increased risk of developing PDAC in the presence of CagA-negative *Helicobacter* strains could be conceivable.

While some groups did not find any correspondent DNA in the pancreatic tissue or juice [31], others isolated DNA in tumor tissue of PDAC patients but not in healthy controls [32]. Interestingly, in this study, the DNA of enteric *Helicobacter* species and *H. pylori* was never present in both the pancreatic and the gastroduodenal tissue. This suggests that migration from the gut into the pancreas seems to be unlikely.

### 3.3. The Bacterial Microbiome of the Pancreatic Tissue and PDAC

The analysis of pancreatic samples and fluids from patients with PDAC compared to samples from patients with a healthy organ or with benign pathologies provided evidence that the pancreas is not a sterile organ [32,33,34,35,36,37,38,39,40,41,42,43,44] (Table 1). In particular, the analysis of pancreatic cystic fluid revealed a specific bacterial ecosystem which may reflect the microbiota harbored within the pancreas [39].

Despite substantial inter-individual variability of the gut flora, some studies concur in their findings, pointing at different bacterial species potentially involved in PDAC tumorigenesis.

The most prominent microbes identified in pancreatic tissue samples and associated with PDAC are Gram-negative bacteria, more specifically from the phylum *Proteobacteria* [35,44]. Pushalkar et al. directly compared pancreatic and fecal samples, showing especially for *Proteobacteria* an increased presence in the pancreas tissue. Among *Proteobacteria* genera, *Pseudomonas* were the most abundant in PDAC [41]. Thomas et al. also detected increased *Proteobacteria* in human PDAC tissue [42] while Chakladar et al. showed an association for some members of the classes *Betaproteobacteria* and *Gammaproteobacteria* with poor patient prognosis [35]. Analogously, Geller et al. showed an elevated presence of *Gammaproteobacteria* in tissue samples of subjects with PDAC [37], however, results of another study showed enrichment of the genus *Pseudoxathomonas* of the classes *Gammaproteobacteria* within the pancreas among PDAC long-term survivors [43]. Of note, elevated levels of *Proteobacteria* were detected in fecal samples of patients with PDAC [41,45], making stool analysis an attractive, cheap, and non-invasive method to detect intrapancreatic microbial shifts.

Some studies showed elevated levels of intratumoral *Enterobacteriaceae*, which also correlated with poor prognosis [35,37]. Moreover, another intestinal bacterium, *Enterococcus faecalis*, was identified within juice and tissue samples of PDAC-patients [40]. Since these bacteria are typical of the human gut, their presence within the pancreas could suggest a translocation from the gut.

*Fusobacterium* spp., a bacterial genus commonly present in the oral cavity during periodontal disease, was also found in PDAC tissue samples. Its presence was independently associated with a worse prognosis. However, so far, no data show an effect of its eradication, and neither genetic nor molecular PDAC patterns showed any correlation with *Fusobacterium* colonization [33,38]. Similarly, *P. gingivalis* has also been detected in significantly higher concentrations within the pancreatic duct of periampullary malignancies [44] and in fluid of pancreatic cysts obtained through endoscopy [39].

Specific microbial signatures have also been associated with cystic precursor lesions and different PDAC tumor stages. Xy et al. observed that high-grade IPMN showed high levels of *Fusobacterium nucleatum* and *Granulicatella adiacens* compared to non-IPMN cystic lesions [38]. Another study showed *Firmicutes* spp. (*Streptococcus* and *Veillonella*) prevalence in stage I/II PDAC, while *Bacteroides*, *Proteobacteria*, and *Synergistetes* were more prominent in stage IV [41]. Of note, *Bacteroides* was also found in higher concentration in PDAC tumor samples [38], in particular the genus *Elizabethkingia* [41]. These data suggest microbiome changes during tumor development.

Some microbiome patterns have also been described to act protectively. In particular, higher α-diversity (an indicator for bacterial variability) and higher levels of *Saccharoplyspora*, *Streptomyces*, and *Pseudoxanthomonas* were associated with PDAC long-term survival [43]. In contrast, anaerobes like *Lactobacillus, Roseburia*, and *Faecalibacterium*, known to exert systemic anti-inflammatory effects, were significantly reduced in PDAC tissue [44].

Curiously, in one study the direct comparison of microbiota analyzed in tissue samples of healthy pancreas, chronic pancreatitis and PDAC did not show any differences between the samples [42].

Despite the large amount of published work, the question about the route by which these microorganisms reach the target organ is still not answered. Even though the oral administration of *Saccharomyces cerevisiae* in mice was followed by a consistent presence of this microorganism within the main pancreatic duct [34], there is so far no proof of ductal migration of specific PDAC associated bacteria in humans.

## 4. The Role of the Microbiome in PDAC

The isolation of bacterial DNA directly from healthy and tumorous pancreatic tissue generates new research perspectives. On the one hand, it encourages identification of specific microbial signatures in the gut as novel non-invasive tumor markers, on the other hand, it adds a fascinating new player in carcinogenesis.

### 4.1. Microbiota and PDAC Induction

Most pancreatic cancers are believed to develop from non-invasive premalignant lesions, histologically defined as pancreatic intraepithelial neoplasia (PanIN). In these lesions, somatic mutations in genes like Kirsten rat sarcoma (KRAS) (codons 12, 13, 61) or, less frequently, guanine nucleotide binding protein (GNAS), are an early, almost universally found event [46]. Even though direct microbiota-associated tumor induction has not been described so far, some microorganisms have been observed to be associated to genetic alterations in PDAC (Figure 1).

*P. gingivalis*, for example, can secrete peptidyl-arginine deaminase, an enzyme that is known to produce point mutations in tumor protein p53 (TP53) and KRAS [50]. *Toxypothrix* sp., *Acidovirax ebreus* and *Shigella sonnei* also correlate with the downregulation of signatures directly related to TP53 [35]. Obesity-induced alterations of gut microbiota could also play a role due to the higher incidence of *Firmicutes* and the reduced numbers of *Bacteroides* [51] which lead to a pronounced production of deoxycholic acid, a bacterial metabolite known to cause point mutations [52].

The presence of *H. pylori* in human pancreatic cells has been associated with higher levels of nuclear factor kappa-light-chain-enhancer of activated B cells (NF-kB), activator protein (AP) 1, interleukin (IL) 8, vascular endothelial growth factor (VEGF), and serum response elements, all factors associated with tumor induction [53]. In this regard, a direct carcinogenic action of *H. pylori* has also been demonstrated in gastric cancer by deregulating polyamine metabolism and promoting oxidative stress [54]. Interestingly, increased serum concentrations of polyamines have also been found in mice and human PDAC subjects. In this case, *Lactobacillus reuteri* was described to be at their origin [55].

Endogenous carcinogens like nitrosamines have also been found in higher concentrations in in vivo models of PDAC. Their origin remains unclear. However, their extrapancreatic source and secondary transport to the target organ via bloodstream suggest a distant located microbial dysbiosis [56].

Of great interest is the recent discovery of epigenetic alterations in PDAC related to *Proteobacteria* (like *Aggregatibacter aphrophilus* and *Agrobacterium radiobacter*), Gram-positive bacteria (like *Beutenbergia cavernae*) and *Mycoplasma hypopneumoniae*. These bacteria were strongly associated with an upregulation of specific methylation-related gene expression signatures [35].

### 4.2. Microbiota and PDAC Progression

Several preclinical models confirm the distant (gut) and local (intrapancreatic) role of microbiota in tumor progression.

Different mechanisms of microbiota-related tumor progression have been proposed over the last decade. The currently two most intensively debated hypotheses are cancer-associated inflammation and pro-tumorigenic immunomodulation within the tumor microenvironment (TME).

#### 4.2.1. Cancer-Associated Inflammation

Oxidative stress disbalance is a pivotal mediator of inflammatory-induced carcinogenesis, and chronic inflammation has been recognized as a central facilitator in pancreatic carcinogenesis [57].

One of the proposed models sees a high-fat diet (HFD) as leading cause of an inflammatory response, which finally results in tumoral development of PanIN [58]. In a mouse model of colorectal carcinoma (CRC) bearing mutated KRAS a major structural change of the gut microbiota was identified as the link between HFD and inflammation [59]. Some authors suggest that within the pancreas, where KRAS mutation alone is insufficient to initiate an invasive carcinoma, the synergistic effect of microbe-induced inflammation and KRAS mutation could sustain the tumorigenic process [60]. Similarly, Gram-negative bacteria colonization of the biliopancreatic tree has been related to a tumor-associated inflammatory status [61]. More specifically, PDAC-cells exposed to *E. faecalis* showed elevated expression of the pro-inflammatory cytokines CXCL8 and VEGF, that are known to promote fibrosis and angiogenesis. Moreover, abundance of *E. faecalis* was found both in samples of PDAC and CP, suggesting a possible role of this bacterium in malignant degeneration of CP [40]. Additionally, colonization of PDAC by *Citrobacter freundii* and *Pseudomonadales bacterium* has been correlated with the upregulation of proinflammatory immune pathways such as the inflammasome [35].

Smoking has been characterized as a main risk factor for PDAC and it is further linked to bacterial dysbiosis [5]. Curiously, elevated levels of *A. baumannii* and *M. hyopneumoniae* found on PDAC samples correlated with smoking-mediated changes in the genome that cause pancreatic cancer [35].

Nutritional habits influence the microbial composition. Beneficial species of the gut flora like *Roseburia* and *Eubacterium rectale* were decreased by low-carbohydrate and high-protein diets [62]. Same dietary regimens are related to reduced intestinal butyrate levels, a short-chain fatty acid involved in cell differentiation, apoptosis, and histone hyperacetylation, all effects thought to be associated with carcinogenic processes [62]. In contrast, high energy diets can cause the activation of pattern recognition receptors (PRRs) like Toll-like receptor type 4 (TLR4) by facilitating the absorption of bacterial lipopolysaccharide (LPS) in the gut. This in turn results in a pro-tumorigenic systemic low-grade inflammation [63] (Figure 1).

These observations suggest a role for intestinal dysbiosis in facilitating an aspecific inflammatory status that has its origin in external factors, like dietary habits, and exerts its effect in the gut and in more distant locations like the pancreas.

#### 4.2.2. Pro-Tumorigenic Immunomodulation of Innate and Adaptive Immunity

Microbial dysbiosis, acting remotely (gut) and locally (intrapancreatic), has been associated with a TME shift towards an immunotolerant phenotype. In particular, Chakladar et al. demonstrated that *C. freundii* and *M. hyopneumoniae* correlate with multiple immunosuppressive pathways [35]. Some studies even consider the microbiota as a new component of the TME [64].

A role of the microbiota acting remotely from the gut has been recently demonstrated in heterotopic mouse xenografts. Thomas et al. successfully abolished PDAC growth after administration of wide-spectrum antibiotics. Bacterial depletion resulted in an increased expression of tumor suppressor genes death-associated protein kinase 2 (DAPK2), Krüppel-like factor 9 (KLF9), and Lumican (LUM) while microbiota-intact mice showed upregulation of pro-tumorigenic genes tenascin C (TNC), chemokine (C-X-C motif) ligand 10 (CXCL10), and plexin-A4 (PLXNA4). The immune status of the TME within the PDAC differed significantly, depending on the presence or absence of the intestinal microbiota. Mice lacking adaptive immune system (non-obese diabetic–severe combined immunodeficiency: NOD-SCID) had increased CD45^+^ innate immune cells in their PDAC xenografts only if treated with wide-spectrum antibiotics. Untreated mice had a lower number of CD45^+^ infiltrates, resulting in increased tumor diameter [42]. These observations suggest an intrapancreatic, microbiota-mediated suppression of the innate immune system and of the immune surveillance.

Other studies suggest that gut microbiota can influence tumor progression by shaping the adaptive immune system. In a heterotopic mouse model of PDAC, the antibiotic-driven depletion of gut bacteria resulted in increased numbers of anti-tumorigenic lymphocytes, like CD3^+^CD4^+^IFNγ^+^, CD3^+^CD8+IFNγ+, CD3^+^IFNγ^+^, and reduced occurrence of pro-tumorigenic CD3^+^IL-17^+^ and CD3^+^CD4^+^IL-10^+^ cells [65]. Similarly, mouse models of slow progressive PDAC (p48^cre^; LSL-Kras^G12D^ known as KC-mice) and PDAC xenografts also showed suppression of the intratumoral adaptive-immunity cells [41,66]. Of note, feces of these KC-mice showed in vitro the ability to drastically reduce the activation of CD4^+^ and CD8^+^ cells, and T_h_1-cells differentiation was decreased as well [41]. In contrast, germ-free KC-mice showed higher intratumoral anti-tumorigenic lymphocytes. However, after stool transplantation from mice harboring an aggressive form of PDAC (Pdx1^cre^; LSL-Kras^G12D^;Tp53^R172H^, known as KPC-mice), anti-tumorigenic lymphocytes were significantly reduced [67].

The molecular pathways leading to these immunologic alterations are still not clear. However, available data suggest an essential role of the TLRs in pancreatic tumorigenesis. TLRs are known to be part of the innate immune system as they initiate the antimicrobial response. Its increased presence on murine and human PDAC tumor cells and macrophages [41,47,68] also suggests a crucial role as intratumoral immunomodulators. In the presence of KRAS mutations, TLR-signaling through TIR-domain-containing adapter-inducing interferon-β (TRIF) activation has been shown to promote PDAC progression [49,68]. However, TLR-signaling has also been described to be involved in pancreatitis [69]. In that context, inhibition of the TLR-associated myeloid differentiation primary response 88 (MyD88) pathway by intrapancreatic dendritic cells (DC) resulted in a pro-tumorigenic, fibroinflammatory environment, with consequent T_h_2-shift and acceleration of the transition from pancreatitis to carcinoma [47].

Activation of TLR2, TLR4, and TLR5 was observed to be higher expressed in mice exposed to a cell-free extract of gut bacteria-derived from KC-mice compared to wild-type (WT) mice [41]. Other studies showed microbiota-induced activation of TLR4 and TLR7 resulting in pro-tumorigenic immunosuppressive TME in early and progredient tumor stages [47,49,70] TLR5 has been described to be upregulated in TME macrophages, and to be related to tumor growth [41]. Finally, TLR9 activation has been demonstrated to induce pancreatic stellate cells (PSCs) of the tumoral stroma to become fibrogenic and to attract T_reg_ cells and myeloid-derived suppressor cells (MDSCs) [49] (Figure 1).

Regarding TLR2, the findings are less clear. While high expression and activation on tumor-associated macrophages (TAM) is related to tumor growth in a PDAC mouse model [41], TLR2-agonists have been shown to be an effective adjuvant immune-therapy against PDAC [71]. Moreover, tumor progression due to intracellular presence of *P. gingivalis* under hypoxic conditions has been demonstrated to be independent from TLR2 signaling. On the contrary, in the context of oral carcinoma *P. gingivalis* promotes tumor growth in a TLR2-dependent manner [17].

Some TLRs have been already addressed as potential targets for immunotherapy in pancreatic cancer. As already reported, synthetic high affinity TLR2 agonists have been observed to induce boost immunity when given as vaccine adjuvants in murine PDAC models [71]. In a phase I/II, trial patients with incompletely resectable PDAC received during surgery an intratumoral injection of MALP-2, a synthetic lipopeptide which activates the immune response through TLR2/6. Combined with an adjuvant gemcitabine administration the authors showed a median survival of 9.3 months, and no distant metastases were reported during the follow-up [72]. Therapeutic strategies targeting TLR7 are still in a preclinical stage producing, however, incongruent results. Antitumoral effects like inhibition of stromal proliferation have been observed in murine PDAC models following stimulation as well as following inhibition of this receptor [68,73].

Taken together, these findings show a complex interaction between microbiota and intrapancreatic immune cells in the context of PDAC. Even though we are far from fully understanding which specific pathway and molecular signaling are involved in establishing this intratumoral immunotolerant phenotype, microbiota have been identified as an important player in this setting. Some authors propose a Janus-faced involvement of TLRs. While the peripancreatic pro-inflammatory response might be only the first effect of TLR-activation, thanks to molecular feedback control mechanisms the same receptors could for a second time modify their signaling and switch from a Th1 to Th2 response with immunosuppressive features [47,74,75] (Figure 1).

### 4.3. Microbiota and PDAC Treatment

The complex interactions between microbiome and immune system also seem to influence prognosis and treatment response to different adjuvant therapies.

Regarding prognosis, microbial ablation has been demonstrated in vivo to significantly decrease the rate of tumor progression from PanIN to PDAC [42]. Concerning the influence of systemic treatments in a human study, the ablation of *Klebsiella pneumoniae*, which may promote gemcitabine resistance in PDAC, has been associated with improved survival [76]. Similarly, bacterial ablation enabled checkpoint inhibitors’ efficacy by upregulating programmed cell death protein 1 (PD-1) expression in PDAC mouse models [41] (Figure 1). Of note, in renal and non-small cell lung carcinoma, bacterial ablation reduced the effectiveness of checkpoint blockade therapy [77]. This apparent contradiction could depend on the type of antibiotics used for the ablation. In fact, broad-spectrum antibiotics could lead to a complete microbiota depletion, including anti-tumorigenic species [44]. Hence, a selective antibiotic ablation combined with other systemic treatments could represent a favorable strategy, especially if one bears in mind that human PDAC tissue has been demonstrated to harbor predominantly Gram-negative bacteria [41,76].

### 4.4. Metabolomics and Proteomics: Novel Functional Approaches in PDAC-Microbiome Studies

While genomic analyses focus on comprehensive bacterial species profiling, new functional tools like proteomics and metabolomics address dysbiosis from another perspective. These studies concentrate their efforts on the harmful protein microenvironment caused by the dysbiosis.

One of the first conducted metabolomic studies focused on the presence of elevated serum polyamine levels in a genetically engineered PDAC murine model (KPC-mice) as well as in human PDAC samples. Polyamines are of known bacterial origin. Interestingly, they emerged in KPC-mice already early before any detectable tumor was located and were also present in human serum samples of PDAC patients. These data suggest early changes in the gut flora in patients developing PDAC and could therefore represent a new, early non-invasive marker [55].

Recently, metabolomics analysis has been also performed on serum and cystic fluids of patients with neoplastic and non-neoplastic cystic lesions of the pancreas. Among them, PDAC precursors, showed significant correlations with defined metabolic patterns, like purine oxidation, heme metabolism, acyl-carnitines and glycolytic metabolites. Furthermore, absolute quantitative measurements on cyst fluid highlighted acyl-carnitines as the top discriminants between neoplastic and non-neoplastic cystic lesions. The observed correlation between 16S RNA copy numbers and metabolite levels stresses the microbial origin of this metabolic “signature” [36].

Of note, proteomic bacterial profiling of bile from patients bearing PDAC showed overexpression of IL-8, which is known to be stimulated by bacterial biofilm formation. The authors of this study also observed elevated levels of primary and secondary compounds, which play an important role in biofilm formation and act as inhibitors for concurrent bacterial species, suggesting the presence of major competition among different bacteria in this context [78].

## 5. The Role of the Mycobiome in PDAC

The role of microbial components other than bacteria in tumorigenesis is quite unexplored [79]. The fungal component is known as “mycobiota”, with the term “mycobiome” indicating their collective genomes. Fungi are estimated to comprise less than 1% of all commensal species [80].

Recently, mycobiome alterations have also been observed in the context of human malignancies like CRC and PDAC, with the intrapancreatic mycobiome of PDAC patients clustering differently from that of healthy individuals [34]. PDAC and CRC showed elevated *Basidiomycota* levels like *Malassezia* spp., while *Ascomycota* were reduced. In particular, *Malassezia globosa* showed good accuracy in differentiating CRC and PDAC from healthy controls [34,81].

Interestingly, *Malassezia* was also the most prevalent genus in the pancreas of KC-mice. In this murine PDAC model, levels increased in parallel with tumor growth, reaching its peak when tumor development was completed. Interestingly, in earlier life stages wild-type and KC mice do not differ in their mycobiome, pointing at fungal dysbiosis and especially at *Malassezia* as a crucial player in PDAC development [34].

### 5.1. Molecular Mechanisms of Mycobiota Related PDAC Development

*Malassezia* is commonly found on the skin with the capability of gut colonization [82]. It encodes some secreted enzymes similar to *Candida albicans*, which have also been described to contribute to carcinogenesis [83,84].

Its contribution to PDAC progression seems to be related to the presence of mannose-binding lectin (MBL). Higher MBL expression was associated with worse survival in PDAC patients. In contrast, in MBL-null mice *Malassezia* did not accelerate tumor progression, nor did the treatment with amphotericin B protect from tumor growth in these mice [34].

#### 5.1.1. The Role of MBL

MBL is a soluble PRR, which recognizes among other pathogenic carbohydrate antigens fungal pathogens and activates the lectin pathway of the complement cascade [85]. This results in the production of C3a. The oncogenic activity of C3a has been demonstrated in mice with tumor growth mitigation due to deficiency of C3 or its receptor, suggesting a crucial role of this pathway in tumor development [34].

A complement-driven tumor progression has also been observed in human specimens [86]. C3a has been classified as potentially oncogenic since it could increase proliferation, motility, and invasiveness of tumor cells. Signaling of complement receptor C3a (C3aR) has been shown to be involved in epithelial–mesenchymal transition (EMT), while C5a acts in an immunosuppressive way by inducing apoptosis of CD8^+^ cytotoxic cells, by attracting MDSC into the tumor, and by participating in the shift of the macrophages towards an M2-phenotype [87]. Furthermore, the expression of C3a was also associated with reduced survival of PDAC patients [34].

Of note, high levels of CD59, an inhibitor of the membrane attack complex (MAC) of the complement, have been observed on pancreatic cancer cells. This could explain why in the context of PDAC only the tumorigenic effects of the complement are present, while the lytic activity of the MAC seems to be suppressed. Moreover, the expression of CD59 appears to be induced by alternatively activated macrophages [88], whose polarization could be influenced by DCs and Dectin-1 activation (Figure 2).

To the best of our knowledge, no treatment targeting the MBL pathway has been described so far in a preclinical and/or clinical PDAC setting.

#### 5.1.2. The Role of Dectin-1

Dectin-1 is another fungal PRR that has been described as an emerging player in pancreatic oncogenesis. Dectin-1 is a non-classical C-type lectin receptor generally expressed on the surface of myeloid-monocytic cells and some T cells that recognize β-1,3 and β-1,6 glucan polysaccharides expressed mostly by yeasts and fungi, with its activation resulting in NF-kB expression [89]. This receptor has been recently observed on PDAC and tumor-infiltrating myeloid cells of human and murine tissues [91].

Dectin-1 is characterized by functional selectivity. Its response varies depending on the binding ligand [92]. Dectin-1 can be activated by fungal glucans leading to the activation of the innate immune system and to the expansion of DCs through GM-CSF production [89]. In addition, Dectin-1 expression on DCs and macrophages is critical for natural killer cells mediated elimination of tumor cells expressing N-glycan structures [94]. On the other hand, the interaction of Dectin-1 with Galectin-9, a lectin with an affinity for β-galactoses highly expressed on PDAC cells, leads to CD8^+^ T cell exhaustion via T-cell immunoglobulin domain and mucin domain 3 (TIM-3) checkpoint receptors [91]. Galectin-9 and its functions have only been recently characterized. It is still unclear whether it represents a “sterile” ligand for Dectin-1, or whether its expression is linked to the fungal colonization of the pancreas. Of note, Galectin-9, independently from its molecular function, has been recently proposed as a possible new serum marker for PDAC [95]. Another study observed Dectin-1 as tolerogenic receptor for annexins, proteins expressed on apoptotic cells, which can induce immune tolerance via nicotinamide adenine dinucleotide phosphate (NADPH) oxidase-2 and can also inhibit the NF-kB pathway [92,93] (Figure 2).

#### 5.1.3. The Crosstalk between Different Fungal PRRs

As already mentioned, TLR2 has been related to PDAC. Fungal presence within PDAC tissue together with elevated expression of Dectin-1 and TLR2 suggests crosstalk between mycobiota and cancer cells. However, the function of TLR2 is not clearly defined. On the one hand, TLR2 activation has been related to tumor growth [41]; on the other hand, TLR2-agonists act antitumorigenic inducing apoptosis on PDAC cells [71].

Despite the Janus-like observations about the isolated Dectin-1 and TLR2 activation, synergisms between these two receptors have been described when binding β-glucan [96] (Figure 2). Since *Malassezia* contains both β-1,3 and β-1,6 glucan, it could be suggested that those two molecules could, respectively, bind Dectin-1 and TLR2, finally leading to their synergistic activation. In a mouse model of type-1 diabetes this synergistic activation resulted in the expression of immunoregulatory cytokines like IL-10 and TGF-β as well as indoleamine 2,3-dioxygenase (IDO). These compounds act synergistically in a tolerogenic way stimulating T_reg_ [90].

TLR9 also needs to be considered in the context of mycobiome related PDAC development, since it has been linked to induction of stroma-producing PSCs, T_reg_ and MDSCs [49] and its expression on macrophages can be stimulated by different fungi, including *Malassezia* [97].

### 5.2. Mycobiota and PDAC Progression

Mycobiome alterations seem also to be related to PDAC progression. In different mouse models, the ablation of *Malassezia globosa* with amphotericin-B was protective against tumor progression. It boosted the tumor-shrinking effect of adjuvant gemcitabine-based chemotherapy too. Of note, the recolonization of amphotericin-B pretreated mice with *M. globosa* accelerated the tumor growth. The same did not happen by recolonization with other fungi, like *Candida* spp., *S. cerevisiae*, and *Aspergillus* spp. Besides, amphotericin B gavage in germ-free, tumor-bearing mice did not influence tumor progression at all [34].

## 6. Limitations and Challenges of Micro- and Mycobiome Studies

Although some studies showed a certain level of agreement in their findings, the general interpretation of data regarding the role of micro- and mycobiome in the context of PDAC is not always consistent. This reflects the challenges this research is faced with.

Notoriously, the major issue concerning microbiology studies is the wide range of possible confounding factors. Oral microbiome studies, for example, are strictly dependent on the method of sampling (saliva vs. tongue collection). Moreover, smoking habits and strong environmental differences between analyzed patient groups also put in perspective these findings. Paradigmatic are the observed differences in studies regarding *Helicobacter* spp., whose specific epidemiologic geographical distribution could lead to consistently different findings regarding its role in pancreatic carcinogenesis [98].

A major challenge represents also the heterogeneity of the gut microbiota itself. Bacteria and fungi live together with viruses, archaea and protozoans. Furthermore, the number of fecal bacteria is much higher compared to that of, e.g., fungi (10^11^ bacteria per gram versus 10^6^ fungi per gram) [99]. However, while the concentration of fungi seems to be stable throughout the entire gastrointestinal tract the number of bacteria increases towards the colon, resulting in different fungi:bacteria ratios along the road [100,101]. Even though not described in the field of carcinogenesis, different interactions between bacteria and fungi [102] like commensalism, competition or even mutualism might have pivotal roles in generating microbial dysbiosis which then culminate in local immunologic alterations and tumor development. These aspects should be considered in future experimental approaches.

Another caveat consists in the avoidance of possible sample contamination. This challenge is addressed by different approaches like the elimination of the external tissue layer of a specimen before analyzing it, or the adjustments of PCR procedure by excluding known contamination sequences. Other methods used to reduce analysis errors due to contamination consist in the comparison of pancreatic samples with negative tissue controls or alternatively the bioinformatic selection of possible result biases (Table 1).

Regarding studies of the pancreatic microbiome, the specimen analyses of normal pancreas and benign or malignant pancreatic diseases have been mostly conducted on small, non-homogeneous patient cohorts. Especially in the cases of malignant specimens, the recruitment of a representative patient cohort is a task hard to overcome. Neoadjuvant chemotherapeutics, the perioperative antibiotic prophylaxis, preoperative biliary stenting (with consequent contamination of the pancreato-biliary tree) are the most important confounding factors that have to be taken into consideration since they are known to profoundly influence the gut microbiome [103,104].

All these issues pose, for the research regarding the influence of the microbiome on PDAC development, tremendous challenges that are currently not uniformly addressed in the literature. Prospective, multicenter studies including large patient groups and reaching agreements about a common strategy to cope with sample contamination might lead the way in providing a clearer picture in this regard.

## 7. Conclusions

Over the last decade, the literature provided increasing evidence that dysbiosis of the intestinal flora plays a role in PDAC development. Despite available descriptions of specific microbiome signatures in feces or the oral cavity of PDAC patients, inter-individual variability of the microbiome hampered its use as a tumor marker.

The observation of intrapancreatic microbiome dysbiosis makes it, however, an appealing treatment target that could be integrated into systemic treatment strategies. This could also be valid for the mycobiome and particularly for *Malassezia globosa*.

The findings that *Proteobacteria,* like *K. pneumoniae*, and fungi, like *Malassezia*, share similar recognition receptors on immune cells, and that their depletion by quinolone [76] and amphotericin B [34], respectively, showed efficacy in preventing tumor progression in mouse models, highlight a critical role of the crosstalk between intrapancreatic flora and TME.

Future studies should focus on these interrelations to identify new treatment targets to improve the abysmal prognosis of PDAC.

## Figures and Tables

**Figure 1 cancers-13-03431-f001:**
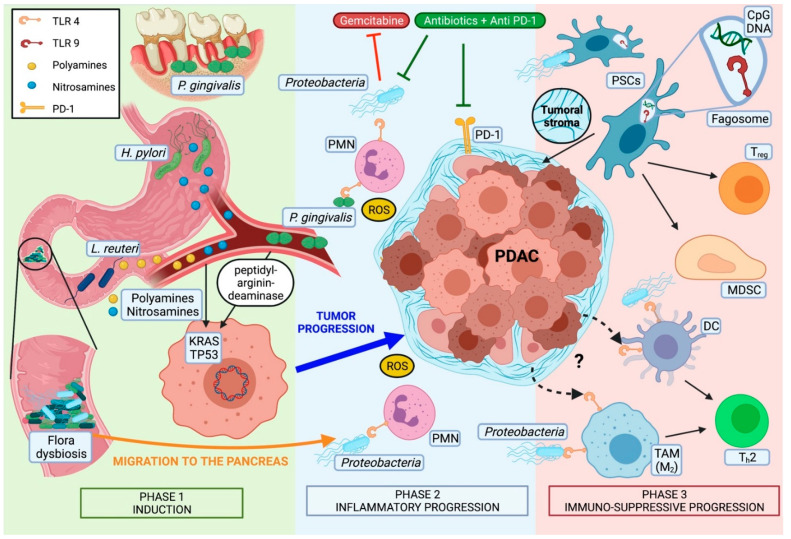
Proposed model of microbial dysbiosis driven pancreatic carcinogenesis. Phase 1: The tumor induction in the case of flora dysbiosis is related to the production of different substances that can be responsible for point mutations of genes like KRAS and TP53 of pancreatic ductal cells (e.g., polyamines produced by *H. pylori* and *L. reuteri*, or the enzyme peptidyl-arginine-deaminase of *P. gingivalis*). Phase 2: Tumor progression after the proliferation of the first cell-clone is sustained by bacterial-induced inflammation. Dysbiosis of the gut flora and alterations of the intestinal wall permeability originate from diet-disbalance and finally facilitate the migration of microorganisms into the pancreas. In particular, the translocation of Gram-negative bacteria elicits an inflammatory response. This one occurs when PMNs recognize bacterial LPS via TLR4 with consequent production of ROS. In this way, the establishment of an oxidative stress disbalance sustains the carcinogenic process. Phase 3: Intrapancreatic mechanisms of receptor-related molecular feedback lead for a second time to a switch of the immune response towards a tolerogenic phenotype. In particular, the activation of TLR4 expressed by DCs and M2-polarized TAM induces T_h_2-deviated CD4^+^ cells. However, it is still unclear if this receptor function of TLR4 depends on binding of either bacterial LPS or other uncharacterized tumoral products (marked with “?” in the picture) [47,48]. Furthermore, the activation of TLR9, an essential receptor for the recognition of CpG bacterial-DNA expressed on PSCs, stimulates the production of fibrous stroma and the expression of CCL11, a mediator with pro-tumorigenic effects on pancreatic ductal cells. TLR9 activation also leads to the PSC-dependent recruitment of T_reg_ and MDSCs in the TME [49]. Abbreviations: DC: dendritic cell; MDSC: myeloid-derived suppressor cell; PMN: polymorphonuclear cells; PSCs: pancreatic stellate cells; ROS: radical oxygen species; TAM (M2): tumor-associated macrophages with M2 polarization; T_h_2: T-helper type 2 cells; T_reg_: T-regulatory cells. (Picture created in BioRender.com, https://biorender.com, accessed on 23 April 2021).

**Figure 2 cancers-13-03431-f002:**
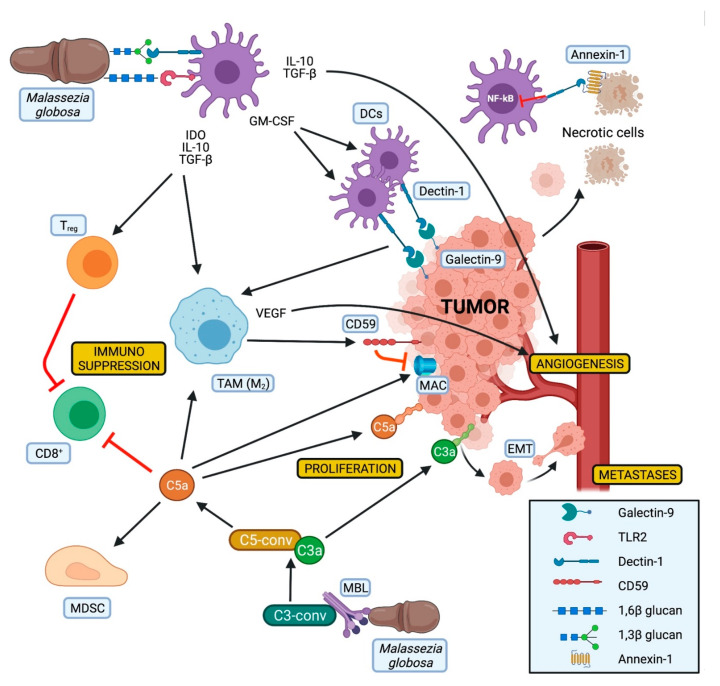
Proposed model of the role of *Malassezia* spp. in pancreatic tumor progression. The relation between *Malassezia* and pancreatic tumor progression is linked to the action of both cellular and molecular effectors, which lead to an intratumoral immune shaping. (1) Dendritic-cells (DCs): the synchronous recognition of fungal antigens like 1,3- and 1,6-β glucan by DCs lead to the production of GM-CSF [89], with consequent cell-expansion and release of indoleamine 2,3-dioxygenase (IDO), IL-10, and TGF-β. These mediators favor the activation of T_reg_, which inhibit T-cells cytotoxicity (CD8^+^) and stimulate a switch of tumor-associated macrophages (TAM) towards an M2-phenotype [90]. On the other hand, IL-10 and TGF-β, together with VEGF produced by the TAMs, stimulate the intratumoral angiogenesis. Thanks to the expansion of DCs, more Dectin-1 can bind Galectin-9, a lectin expressed on tumor cell membrane, and contribute to the M2-shift of the TAM [91]. Moreover, Dectin-1 can bind Annexin-1 on dying tumoral cells, leading to NF-kB inactivation [92,93]. (2) Complement system: the recognition of *Malassezia* through MBL activates the complement cascade, leading to formation of active C3 and C5 convertases. Among the different complement components, C3a and C5a both lead to tumor cell proliferation by binding their specific receptors expressed on PDAC cells. Furthermore, the signaling of C3aR increases the epithelial–mesenchymal transition (EMT), promoting the metastatic process. C5a acts in an immunosuppressive way by inducing apoptosis of CD8^+^ cytotoxic cells, attracting MDSC into the tumor, and participating in the shift of the macrophages towards an M2-phenotype [87]. Of note, the fact that TAMs can induce the expression of CD59 on PDAC cells limits the antitumoral activity of the MAC. Abbreviations: C3-conv: C3-convertase; C5-conv: C5-convertase; CD8^+^: cytotoxic CD8^+^ T-cells; DC: dendritic cell; EMT: epithelial-mesenchymal transition; IDO: indoleamine 2,3-dioxygenase; MAC: membrane attack complex of the complement system; MBL: mannose binding lectin; MDSC: myeloid-derived suppressor cell; TAM: tumor associated macrophages M2 polarized; T_reg_: T-helper regulatory cell; VEGF: vascular endothelial growth factor. (Picture created in BioRender.com, https://biorender.com, accessed on 23 April 2021).

**Table 1 cancers-13-03431-t001:** Human micro- and mycobiota on pancreatic tissue and fluid samples.

Reference	Detection Method(s)	Type and Size of Pancreas Sample	Contamination Evaluation	Findings	Conclusion
Nilsson et al. [32], 2006, Sweden	16S rRNA PCR	40 PDAC tissue 14 NET ^1^ tissue 8 MEN1 ^2^ tissue 5 CP ^3^ tissue 10 benign diseases 7 normal tissue	PCR contamination prevention	75% PDAC, 60% CP positive for *Helicobacter* DNA. Benign and healthy negative	Possible role of *Helicobacter* in CP and PDAC development
Mitsuhashi et al. [33], Japan, 2015	16S rRNA PCR qPCR	302 PDAC tissue 25 normal tissue	Not available	8.8% PDAC positive for *Fusobacterium* sp.	*Fusobacterium* correlates with worse PDAC prognosis
Geller et al. [37], USA, 2017	16S rRNA PCR	20 normal tissue 113 PDAC tissue	Negative control sample	*Enterobacteriaceae* and *Pseudomonas* prevalent in PDAC	Bacteria are a component of the PDAC tumor microenvironment
Rogers et al. [38], USA, 2017	16S rRNA PCR qPCR	50 PDAC tissue	PCR contamination prevention	PDAC enriched with *Klebsiella* and *Acinetobacter*	Bacteria are a component of the PDAC tumor microenvironment
Li et al. [39], The Nedetherlands, 2017	16S rRNA PCR NGS ^4^	69 pancreatic cystic fluid	Extrapancreatic control sample (duodenum), Bioinformtaic tools	*Bacteroides* spp., *Enterobacteriaceae*, *Acidaminococcus* spp. prevalent in cystic fluid	Pancreatic cysts harbor a specific bacterial ecosystem with possible role in the neoplastic process
Maekawa et al. [40], Japan, 2018	16S rRNA PCR	5 PDAC tissue 20 PDAC juice	Not available	PDAC juice and tissue samples mostly positive for *Enterococcus faecalis*	Possible role of *E. faecalis* in CP and PDAC development
Pushalkar et al. [41], USA, 2018	16S rRNA PCR qPCR FISH ^5^	12 PDAC tissue	PCR contamination prevention	*Proteobacteria*, *Bacteroidetes* and *Firmicutes* prevalent in PDAC	Bacteria are a component of the PDAC tumor microenvironment
Riquelme et al. [43], USA, 2019	16S rRNA PCR rRNA FISH	68 PDAC tissue	PCR contamination prevention, bioinformatic tools	*Proteobacteria Actinobacteria* and *Bacillus clausii* correlate with PDAC long-term survivors	Microbiome diversity determines the survival of PDAC patients
Del Castillo et al. [44], USA, 2019	16S rRNA PCR	51 PDAC tissue 18 CP tissue 8 other (bile duct, small bowel diseases) 34 normal tissue	Physical specimen manipulation	*Proteobacteria*, *Firmicutes Bacteroides*, *Fusobacteria* and *Actinobacteria* prevalent in PDAC. *Lactobacillus* in non-cancer subjects	Different microbiome composition between PDAC and normal pancreas
Aykut et al. [34], USA, 2019	18S rRNA PCR FISH 18S ITS ^6^ sequencing	13 PDAC tissue	Negative control sample	*Ascomycota* and *Basidiomycota* phyla and *Malassezia* genus prevalent in PDAC	Fungi are a component of the PDAC tumor microenvironment
Chakladar et al. [35], USA, 2020	16S rRNA PCR	187 PDAC tissue	Bioinformatic tools	*Proteobacteria* prevalent in PDAC. *Pseudomonadales Acidovorax ebreus C. freundii. S. sonnei* related to worse prognosis *A. baumannii* and *M. hypopneumoniae* correlate with smoke-related PDAC *A. ebreus, C. baumannii and G. kaustophilus* and *E. coli* prevalent in male PDAC	Corroboration of previous results. 13 microbes correlated to the dysregulation of gene signatures related to oncogenic methylation, cancer progression and immune system modulation
Morgell et al. [36], Sweden, 2021	16S rRNA PCR	Cystic fluid from 5 SCN ^7^ 29 LGD ^8^-IPMN 8 HGD ^9^-IPMN 15 IPMN with associated PDAC	PCR contamination prevention	*Firmibutes, Proteobacteria, and Actinobacteria* most common bacteria within pancreatic cystic fluid	Corroboration of previous results on pancreatic cystic fluid. Metabolomic characterization

^1^ Neuroendocrine tumor; ^2^ multiple endocrine neoplasia type 1; ^3^ chronic pancreatitis; ^4^ next generation sequencing; ^5^ fluorescence in-situ hybridization; ^6^ internal transcribed spacer; ^7^ serous cystic neoplasia; ^8^ low-grade dysplasia; ^9^ high-grade dysplasia.

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
