# Peer review of "Micro- and Mycobiota Dysbiosis in Pancreatic Ductal Adenocarcinoma Development"

_cancers, 2021, doi:10.3390/cancers13143431_

Round 1

Reviewer 1 Report

Although this manuscript shows some overlap with the recently published article by Arsinijevic et al (Cancers 2021), it also forms a nice complement of it and both manuscripts together give the reader a complete overview of the current knowledge on this topic.

The manuscript is well written and particularly well illustrated. The limitations of the cited studies are clearly explained, putting all findings into the right perspective.

I can only congratulate the authors for this extensive work and have no suggestions for further improvement of it.

Author Response

We really appreciate the positive feedback. This boosts our motivation to further put our efforts in this research field.

We carefully read the manuscript for spell checks and rephrased a few passages in order to improve the readability.

Reviewer 2 Report

Interesting and well written article. Methods are appropriate. Results are clearly presented. Discussion is interesting.

Author Response

We really appreciate the positive feedback. This boosts our motivation to further put our efforts into this research field.

We carefully read the manuscript for spell checks and rephrased a few passages in order to improve the readability.

Reviewer 3 Report

This is a very well-written and comprehensive review. Congratulations!

May I suggest to expand a bit more on potentially more specific treatment options targeting the identified downstream mechanisms, e.g. MBL, TLR etc. 

Author Response

We are grateful for the positive comments.

In the revised manuscript, we included a paragraph dealing with TLR-targeted treatments. To the best of our knowledge, no data about targeting MBL are available in the PDAC field.

(line 277 – 287; line 416-417)

We carefully edited the manuscript for spell checks.

Reviewer 4 Report

The authors have conducted a systemic literature search to demonstrate the contribution of micro/myco-biota on PDAC development. The manuscript has a broad range of information related to the title. The models used in the manuscript nicely summarize the microbial dysbiosis driven pancreatic carcinogenesis.  However, authors could consider following aspects to strengthen this manuscript.

  1. Most of the studies surveyed in the manuscript used genomic approaches (e.g. 16S rRNA), which identify the taxonomy of the micro/myco-biota profiles. There appeared to be a lack of review on the studies that use the emerging functional approaches (e.g. proteomics) or/and system biology to address the functional impacts of the microbial dysbiosis on PDAC.
  2. In section 5.1, the authors could elaborate more on how fungi-mediated activation of competent cascade contributes to the immune responses (although this was summarized in Figure 2 legend) in cancer progression context.
  3. In TME, it is likely that, bacteria and fungi synergistically promotes the PDAC. So authors may consider to further address this point in the manuscript.

Minor:

  1. Please check scientific naming of microorganisms throughout the manuscript. (Some are italicized, some are not). Also please check the rules of scientific naming of Phylum and Classes (e.g. Proteobacteria and Betaproteobacteria)
  2. Line 176: If an organism’s both gene and species names are not mentioned prior to this line, it is better to mention the full scientific name here. (Saccharomyces cerevisiae)
  3. Section 3.1 and 5.1 could be better organized to improve the clarity and readability.
    1. Section 3.1 could be organized with sub headings, e.g. Translocation of bacteria to pancreas, microbial signatures in PDAC stages/survival…, etc.
    2. Section 5.1 could be organized with sub headings, e.g. The role of MBL..., Other fungal PRR on cancer progression., etc.

Author Response

We appreciate the subjects brought up by the 4th reviewer and we addressed all the issues in the new version.

Ad 1) We included in the re-submitted manuscript a new paragraph focussing on recent studies with metabolomic and proteomic approaches. (line 343 – 368)

Ad 2) We transferred more information from the legend of Fig. 2 in the main text in order to have better readability in the section dealing with fungi-complement interactions. (see section 5.1.1)

Ad 3) We agree with the reviewer and see potential interactions of fungi and bacteria as one of the major challenges in this research field. We added a new paragraph addressing this issue. (line 512 – 522)

Minor issues:

Ad 1 and 2) (line 87, 90, 126, 459, 494, 497)

Ad 3) 

Section 5.1 was reorganized as suggested by the reviewer. (see section 5.1)

Regarding section 3.1. our opinion diverges from the reviewer. Therefore, we decided not to rearrange the section. We hope that the reviewer will be indulgent with regard to this decision.

In addition to the changes mentioned above we carefully read the manuscript for spell checks and rephrased a few passages in order to improve the readability.